# Complex Exercise Improves Anti-Inflammatory and Anabolic Effects in Osteoarthritis-Induced Sarcopenia in Elderly Women

**DOI:** 10.3390/healthcare9060711

**Published:** 2021-06-10

**Authors:** Jaeyong Park, Jongjin Bae, Jungchul Lee

**Affiliations:** 1Institute of Sports Health Science, Sunmoon University, 70, Sunmoon-ro 221 beon-gil, Tangjeong-Myeon, Asan-si 31460, Korea; 2006076@sunmoon.ac.kr; 2Data Center, Hannam University, 70 Hannamro, Daedeok-Gu, Daejeon 34430, Korea; bjj008@hnu.kr; 3Department of Exercise Prescription, Dongshin University, 185, Geonjae-Ro, Naju, Jeonnam 58245, Korea

**Keywords:** sarcopenia, osteoarthritis, complex exercise, myokine, TCSA

## Abstract

We investigated the effects of a 15-week complex exercise program on osteoarthritis and sarcopenia by analyzing anabolic effects and the impact on the activities of daily living (ADLs). Nineteen women aged ≥60 years with sarcopenia (SEG, n = 9) or diagnosed with osteoarthritis with sarcopenia (OSEG, n = 10) were enrolled and underwent an exercise program. Insulin-like growth factor 1 (IGF-1), irisin, myostatin, interleukin-10 (IL-10), and tumor necrosis factor alpha (TNF-a) levels were analyzed pre- and post-intervention. Thigh cross-sectional area (TCSA) was measured pre- and post-intervention via computed tomography. Western Ontario and McMaster Universities Osteoarthritis Index (WOMAC) and Short Physical Performance Battery (SPBB) were assessed pre- and post-interventions to assess ADL. There was a significant interaction effect between SEG and OSEG at the IGF-1 level post-intervention. Irisin increased and myostatin decreased post-intervention in both groups. IL-10 increased and TNF-α decreased post-intervention with a significant interaction effect in the OSEG group. TCSAs increased post-intervention in both groups. There was a significant interaction between the two groups. OSEG showed a greater WOMAC decrease and SPPB increase post-intervention, and there was a significant interaction effect. Combined exercise may be effective in improving biochemical factors, anabolic effects, and ADL in elderly women with osteoarthritis and sarcopenia.

## 1. Introduction

Sarcopenia is caused by the interaction of a variety of factors and pathways, such as environmental factors, endocrine dysregulation, loss of motor neurons, activation of inflammatory pathways, and reduced satellite cell counts [1]. Inflammatory markers are involved in age-related loss of muscle mass. Elevated TNF-α levels increase the anabolic effects on muscles by inhibiting the rapamycin signaling, and inflammatory cytokines have an antagonistic effect on muscle synthesis by diminishing circulating and muscular insulin growth factor (IGF)-1 levels [2]. Interleukin (IL)-10 concentration in the blood is known to inhibit tumor necrosis factor (TNF)-α, and increases in cytokine levels as a result of aging are believed to be closely associated with sarcopenia [3]. Myostatin (Mstn), an extracellular cytokine, is a transforming growth factor (TGF) that hinders the regulation of skeletal muscle mass and growth. Mstn is exclusively expressed in skeletal muscles, where it inhibits the differentiation and proliferation of myoblasts through an antagonistic effect on IGF-1 or follistatin [4]. In addition, Mstn is associated with aging. Yarasheski et al. [5] reported that physically frail elderly women showed the highest elevation of serum Mstn levels and that elevation of Mstn levels is negatively correlated with skeletal muscle mass; however, aging-related changes in circulating or muscle Mstn-immunoreactive protein levels or Mstn mRNA have not been proven [6]. Further research is needed to address the contradictory results pertaining to the relationship between Mstn and aging.

Irisin, a myokine associated with increased muscle mass, is significantly and positively correlated with skeletal muscle mass and strength; therefore, it has been proposed as a potential biomarker of sarcopenia and muscular impairment [7]. Regular exercise has been reported to increase irisin levels and inhibit Mstn, which hinders muscle growth [8]. Ultimately, achieving sufficient muscle mass is important to increase muscle strength, as muscle mass and strength are positively correlated [9].

Studies on the effects of exercise on increasing muscle mass have reported that aerobic exercise facilitates skeletal muscle growth as well as mitochondrial ATP production and improves aerobic capacity, metabolic regulation, and cardiovascular functions [10]. In other words, aerobic exercise restores mitochondrial metabolism, reduces the expression of catabolic genes, and increases muscle protein synthesis [10,11,12]. In addition, exercise is helpful in maintaining the expression of autophagy proteins and may also upregulate the expression of autophagy-related proteins in skeletal muscle [13]. Furthermore, aerobic exercise has been proposed to regulate Mtsn mRNA expression [14].

In the elderly, strength training stimulates growth hormone secretion that promotes muscle cell growth and muscle strength [15]. Strength training also contributes to an increase in bone density [16]. In particular, resistance exercise can restore a significant portion of lost muscle function and delay deterioration of muscle structure associated with aging [17]. In addition, the cross-sectional area of the muscles of the elderly who perform resistance exercise has been reported to increase similarly to that of younger individuals [18].

Osteoarthritis (OA) accompanied by sarcopenia is a chronic disease in which gradual deterioration of joint cartilage causes subchondral damage and secondarily induces inflammation in other tissues, and ultimately impairs physical functioning [19]. 

Previous studies have shown that aging-induced sarcopenia is closely associated with OA [20,21]. The classic feature of OA is pathological inflammation, and cytokines, such as TNF-α and IL-6, play an important role in the destruction of cartilage [22]. Therefore, reducing TNF-α levels is crucial in treating chronic pain caused by OA.

Prolonged drug therapy to treat inflammation may likely lead to adverse reactions, such as infection, autoimmune diseases, and tumors [23]. However, exercise increases the secretion of the anti-inflammatory cytokine IL-10 and inhibits the production of proinflammatory cytokines, such as TNF-α, without any adverse reactions [24]. 

Because cytokines characteristically expressed in sarcopenia may induce OA, it has been suggested that exercise may have protective effects on muscles. Resistance exercise, a classic exercise for OA, is an important intervention used to inhibit muscle loss by facilitating muscle protein synthesis and muscle growth. Regular resistance exercise enhances exercise performance by enlarging muscle fibers and increasing cross-sectional surface area, however it has been reported that there are limitations in promoting mitochondrial protein expression and improving functioning with resistance exercise alone in elderly individuals with sarcopenia [25]. Furthermore, simple resistance exercise may be less effective in older adults because of low levels of mammalian target of rapamycin (mTOR) signaling, which is involved in muscle protein synthesis [26].

Therefore, a single form of exercise is not an optimal therapeutic exercise intervention for aging-related sarcopenia. A balanced combined exercise program including both aerobic and resistance exercises should be recommended to improve aging-induced sarcopenia [27].

Therefore, this study aimed to investigate the therapeutic effects of a combined exercise program on sarcopenia and OA by analyzing its anabolic effects on muscles and activities of daily living (ADL).

## 2. Materials and Methods

### 2.1. Subjects

Elderly women aged ≥60 years who resided in Asan-si were enrolled; of these, 19 women had sarcopenia, defined by muscle strength ≤20 kg based on grip strength [28]. The participants were divided into the no OA group (SEG, n = 9) and a group diagnosed with OA (Kellgren-Lawrence Grade I or II) by an orthopedic surgeon (OSEG, n = 10). We excluded those with a history of musculoskeletal injury in the past 6 months, history of knee surgery or intra-articular steroid administration, and inability to ambulate independently. Before enrollment, subjects confirmed understanding of the purpose of the study and indicated willingness to participate, after being informed of the experimental procedures in detail. This study was approved by the institutional review board of Sunmoon University (SM-201908-045-1). The participants’ physical characteristics are shown in Table 1.

### 2.2. Experimental Design

The experimental procedure of this study is shown in Figure 1.

### 2.3. Measurement of Cross-Sectional Area of the Thigh

The anatomical cross-sectional area (CSA) of the quadriceps muscle was assessed by computed tomography (CT) scanning (Siemens, Munich, Germany), as described previously [29]. With the subject in the supine position, the right mid-thigh CSA was measured by CT at the midpoint from the inguinal crease to the proximal pole of the patella (Figure 2). Briefly, the scanning characteristics were as follows: 120 kV, 300 mA, rotation time of 0.75 s, and field of view of 500 mm. With the subjects lying supine, legs extended, and their feet secured, a 3-mm-thick axial image was taken 15-cm proximal to the base of the patella. The muscle area of the right leg was selected between 0 and 100 Hounsfield units [30], after which the quadriceps muscle was manually traced using ImageJ software (version 1.45 d; National Institutes of Health, Bethesda, MD, USA) [31].

### 2.4. Blood Sampling and Biochemical Analysis

Blood samples were collected by a medical doctor from each participant after 12 h of fasting. The samples (10 mL) were centrifuged (1000× *g*, 4 °C, 15 min) and the serum was aliquoted and stored at −80 °C for subsequent analysis. For the analysis of serum IGF-1, Mstn, TNF-α, and IL-10, 50-, 50-, 50-, and 200-μL of serum samples were analyzed using commercial enzyme-linked immunosorbent assay (ELISA) kits (R&D Systems, Minneapolis, MN, USA). The lower limits of detection were 0.094 ng/mL, 31.3 pg/mL, 15.6 pg/mL, and 7.8 pg/mL, respectively. Irisin (in a 100-μL of serum sample) was analyzed using a commercial kit (Aviscera Bioscience Inc., Santa Clara, CA, USA), with a lower limit of detection of 0.8 mL. All factors were measured using an ELISA reader (Benchmark Plus Microplate Spectrophotometer, Bio-Rad Hercules, CA, USA) at 450 nm. Values below this limit were assumed to be zero for the statistical analysis. The inter- and intra-assay coefficients of variance were <10%.

### 2.5. Complex Exercise Program

The 15-week complex exercise program supplemented Park and Song’s [32] exercise program for the elderly and was conducted by a professional health and exercise instructor at a public health facility. The exercise program implemented in this study is shown in Table 2 and Table 3.

### 2.6. Data Processing

The mean and standard deviation (SD) of each study parameter were computed using IBM SPSS Statistics version 26 (IBM Corp., Armonk, NY, USA). The group and interaction effects for each variable were analyzed using two-way repeated measures ANOVA. Unpaired *t*-tests were performed as post-hoc analysis for differences between groups, and paired *t*-tests were performed for differences between time points. The statistical significance (α) was set at 0.05.

## 3. Results

Table 4 shows the changes in serum myokines (IGF-1, irisin, Mstn, IL-10, and TNF-α) before and after the intervention between the two groups.

IGF-1 significantly increased after exercise in the OSEG (*p* < 0.001). In particular, there was an interaction effect between the two groups (*p* < 0.01). In other words, while the two groups presented similar IGF-1 levels before the exercise, IGF-1 expression markedly increased in the OSEG after the exercise. Conversely, the SEG showed little change in IGF-1 levels. Irisin significantly increased after the intervention in both groups, with no significant differences between the two groups (*p* < 0.01). Mstn significantly decreased in both groups (*p* < 0.01). Although the interaction effect was not statistically significant, as OSEG showed a relatively high variability, the OSEG tended to show a greater reduction.

IL-10 levels increased by 67.52 pg/mL in the SEG, but increased by 130.26 pg/mL in the OSEG, from 101.93 ± 40.83 pg/mL at baseline to 232.20 ± 74.85 pg/mL after the intervention, confirming an interaction effect (*p* < 0.05). In other words, while the OSEG had relatively low levels of IL-10 compared to the SEG at baseline, the levels rose to a level similar to those of SEG after the intervention. There was a group and time interaction effect for TNF-α, where the OSEG showed a statistically significantly greater reduction compared to the SEG (*p* < 0.05). In other words, TNF-α, a characteristic biomarker, was reduced more dramatically in the OSEG.

Table 5 shows the changes in TCSA (cm²) and ADL after the exercise program in both groups.

Both the left and right TCSA (cm²) increased after exercise in both groups. In particular, the SEG showed a greater increase in the left TCSA compared to the OSEG, showing a significant group interaction effect (*p* < 0.01). Both groups exhibited enlarged muscle fibers after the exercise.

The Western Ontario and McMaster Universities Osteoarthritis Index (WOMAC), which measures ADL associated with OA, did not change in the SEG but statistically significantly improved after the intervention in the OSEG (*p* < 0.001), showing an interaction effect between the two groups (*p* < 0.05). In particular, OA-related signs were markedly reduced in the OSEG after exercise. Furthermore, the Short Physical Performance Battery (SPPB), which comprehensively assesses ADL, significantly improved after the exercise program in both groups (*p* < 0.01). There was a significant interaction effect, as the exercise program was more effective in the OSEG than in the SEG (*p* < 0.05).

## 4. Discussion

This study investigated the positive effects of exercise in elderly women with sarcopenia and OA by examining changes in their muscle cross-sectional surface area and physiological parameters, such as blood markers related to myokines, after the exercise intervention. 

In our study, the exercise program significantly increased the levels of IGF-1, irisin, and anti-inflammatory factor IL-10. In contrast, it significantly reduced the Mstn levels, which hinders muscle protein synthesis and the inflammatory marker TNF-α. The WOMAC score, which is related to ADL, significantly changed in the OSEG. SPPB improved in both groups after the exercise.

Aging deteriorates physiological functioning and results in physical changes, marked by diminished muscle functioning and physical fitness. One notable problem is the presence of sarcopenia. From a physiological perspective, aging-induced sarcopenia cannot be neglected because it induces systemic inflammation and results in the secretion of various proinflammatory cytokines.

Muscles are enlarged when there is more signaling for muscle protein synthesis than that for protein degradation, and factors that induce muscle enlargement, which are expressed due to positive overload, such as exercise, activate downstream proteins in myocytes via the PI3K/Akt signaling pathway, thereby inducing protein synthesis, and increasing muscle mass [33]. IGF-1 is involved in cellular proliferation and differentiation, and thus plays an important role in regulating metabolism and growth of tissues, such as muscles, cartilage, and bone, and IGF-1 concentration in the blood and muscles is reduced with aging-related muscle loss [34].

The moderate-intensity exercise used in this study had significant effects on blood myokine concentrations, such as IGF-1, irisin, and IL-10. Muscle training increases the frequency and amount of growth hormone secretion, which in turn promotes the expression of IGF-1 and induces myocyte growth and enlargement, thereby increasing muscle mass and strength [35]. This is in line with previous findings indicating that protein synthesis strengthens the muscular cross-sectional area and musculoskeletal structures and contributes to the activation of the energy system and neural transmission [36].

Mstn is a blood marker associated with the inhibition of muscle growth. Mstn is a member of the TGF-β superfamily, and it acts directly on myocytes to inhibit myogenesis and differentiation of myocytes [8]. Miyake et al. [37] reported that engaging in 30 min of exercise every day for 10 days increased muscle mass and reduced Mstn expression. High-intensity resistance exercise, which is effective for muscle growth has been reported to reduce Mstn expression [38].

In this study, both groups showed reduced Mstn levels after exercise, with a greater reduction in OSEG. Furthermore, TNF-α, an inflammatory marker that hinders muscle protein synthesis, was also significantly decreased. Both groups showed improvements in TCSA after 15 weeks of exercise compared to baseline. Irisin is a hormone released by myocytes that plays an important role in muscle growth by regulating the balance between exercise and metabolic homeostasis, and its levels in the body are positively correlated with engagement in exercise and intensity of exercise [8,39]. Tsuchiya et al. [40] reported that high-intensity exercise increased irisin levels in the blood. This is in line with our results, where both groups showed significant increases in irisin levels and significant reductions in Mstn levels. 

Myokines, which are expressed in skeletal muscles during muscular contractions, are known to directly affect muscle cells (autocrine effect), surrounding tissues (paracrine effect), or other tissues through the blood (endocrine effect) as proteins and hormones released from muscle fibers [41,42]. 

IL-10 secreted in the muscles is a classic anti-inflammatory cytokine that inhibits immune and inflammatory responses; it is released from T-helper cells, monocytes, and macrophages and regulates anti-inflammatory reactions and neutrophils activitys. Moreover, IL-10 is known to inhibit TNF-α-induced inflammatory responses in monocytes [43]. IL-10 rapidly increases in the plasma during exercise, and the positive correlation with exercise intensity clearly suggests that aerobic-anaerobic combined exercise increases the expression of myokines through various physiological mechanisms and supports the spike of myokine expression during high-intensity aerobic exercise and resistance exercise. Inflammatory markers that predict the risk of OA include C-reactive protein (CRP), IL-6, and TNF-α, and when inflammation is induced, proinflammatory cytokines derived from monocytes, macrophages, and adipocytes facilitate the reaction [44]. The systemic inflammatory response induced by muscle loss, in turn serves as a cause of OA as proposed by Benatti et al. [45], and IL-10 is an important anti-inflammatory myokine to be analyzed in this context. 

Furthermore, TNF-α is a marker related to adipocytes, and it is not only involved in the progression of OA but also influences the immune system [46]. However, reducing body weight and body fat through regular exercise decreases TNF-α concentration and enhances inflammation and insulin sensitivity, thereby lowering the risk of cardiovascular disease and metabolic disorders [47]. There are varying reports on the effects of exercise on TNF-α levels. While some studies have reported that moderate-intensity exercise either reduced TNF-α concentrations or did not lead to any changes [48], other studies have reported that high-intensity exercise increased TNF-α concentration [49]. In addition, a study that examined the effects of 7 months of jogging, walking, stretching, jump rope, and cycling in obese women reported that their TNF-α concentration significantly declined [50]. Another study reported that regular exercise decreased proinflammatory cytokines, and that 6 months of moderate-intensity exercise decreased TNF-α concentration in patients with insulin resistance [51]. 

In our study, IL-10 expression induced via TNF-α stimulation was significantly improved after exercise in both groups. In particular, post-exercise expression of proinflammatory and anti-inflammatory cytokines was antagonistic, and there was a significant interaction effect between the two groups. This is consistent with previous findings, and it seems that increased IL-10 concentration induced by exercise continuously inhibited inflammatory factors, such as TNF-α, in older adults with sarcopenia and OA with high inflammatory markers. In particular, a statistically significant reduction in the group with sarcopenia and OA was observed, which was in line with previous findings indicating that moderate-intensity exercise exerts anti-inflammatory effects by reducing TNF-α [52]; thus, moderate-intensity exercise may be highly effective in older adults. However, establishing the optimal exercise intensity is extremely important, as vigorous exercise can cause inflammatory responses and increase blood TNF-α concentration [49].

Sarcopenia and OA not only cause motor symptoms, such as muscle weakness, pain, and reduced balance, but also hinder basic activities, such as gait, stair climbing, sitting to standing, and turning [53]. Older adults with these conditions have restrictions in their ADL, which in turn deteriorates their quality of life. 

In a study on older adults, Stenholm et al. [54] reported that grip strength was significantly reduced in the obesity plus sarcopenia group and the sarcopenia group compared to the normal group among older adults aged 65–79 years and that grip strength was diminished in all older adults aged ≥80 years, regardless of group. Similarly, Chang et al. [7] reported that grip strength was significantly lower in the group with obesity and sarcopenia and in the group with sarcopenia than in the normal group, and that it was the lowest in the group with obesity and sarcopenia. Considering that a similar aging-related inflammatory mechanism is involved in obesity and OA, grip strength measured at baseline among elderly women with an average age of 75 years in this study was low in both the sarcopenia and sarcopenia and OA groups. In other words, while both groups presented significant inflammation at baseline, the sarcopenia and OA group showed an overall improvement in muscle mass, grip strength, and physical functions after the exercise program. These results support the results of many previous studies. In addition, the reduced muscle strength in the sarcopenia group can be partly explained by the results of Deschenes [55] (2004), where the size and number of fast-twitch muscle fibers are reduced in older adults with reduced muscle mass, resulting in reduced muscle strength.

The SPPB was developed in a multi-center study known as the Established Population for Epidemiologic Studies of the Elderly (EPESE) conducted by the National Institute of Aging (NIA) [56]. This tool consists of three categories that are useful and can be easily applied in older adults with sarcopenia and OA. In the present study, we used this tool as a key assessment tool for factors related to ADL. WOMAC significantly changed in the group with sarcopenia and OA, while SPPB improved after exercise in both groups. In particular, the WOMAC score was significantly improved in the OSEG, with significant differences between the two groups. This suggests that aerobic-anaerobic combined exercise may be effective in early OA. In the present study, the WOMAC score was significantly improved after the 15-week combined exercise in both groups.

These results support the effectiveness of the circuit exercise program consisting of aerobic and anaerobic exercises proposed by Takeshima et al. [27] and are in line with the results of Jung et al. [57], in which a 12-week circuit exercise program improved ambulation, balance, and isokinetic muscle function in elderly women with sarcopenia. Further, Gudlaugsson et al. [58] emphasized the importance of a “combined training intervention” based on the results of his study, which showed that endurance performance, such as 6-min walking ability, improved after 6 months of complex exercise in 117 elderly subjects. Thus, a variety of tools should be used to assess older adults’ ADL and quality of life. In addition to SPPB, the Instrumental ADLs and Tinetti Performance Oriented Mobility Assessment are appropriate assessment tools for older adults. 

The complex exercise program proposed in this study was effective in enhancing biochemical and morphological factors as well as ADL in elderly women with sarcopenia accompanied by OA. Thus, the findings of this study provide useful information for elderly women who intend to engage in exercise to prevent sarcopenia. However, since sarcopenia and sarcopenia with OA may be affected by factors other than exercise, qualitative and quantitative follow-up studies on the improvement of muscle function are required

## 5. Conclusions

The 15-week combined exercise program positively changed physiological and morphological factors, such as muscle growth factors and inflammatory factors, and improved the quality of ADL in elderly women with sarcopenia as well as elderly women with sarcopenia accompanied by OA. Thus, exercise programs should be strongly recommended for elderly women to promote healthy and desirable aging.

## Figures and Tables

**Figure 1 healthcare-09-00711-f001:**
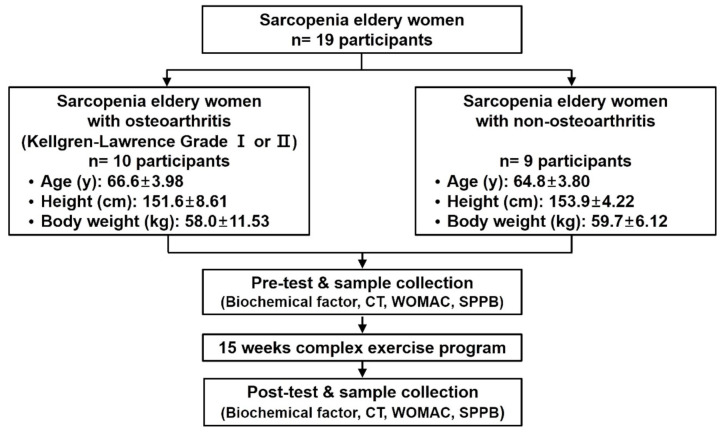
Inspection processes and procedures. WOMAC, Western Ontario and McMaster Universities Osteoarthritis Index; SPPB, Short Physical Performance Battery.

**Figure 2 healthcare-09-00711-f002:**
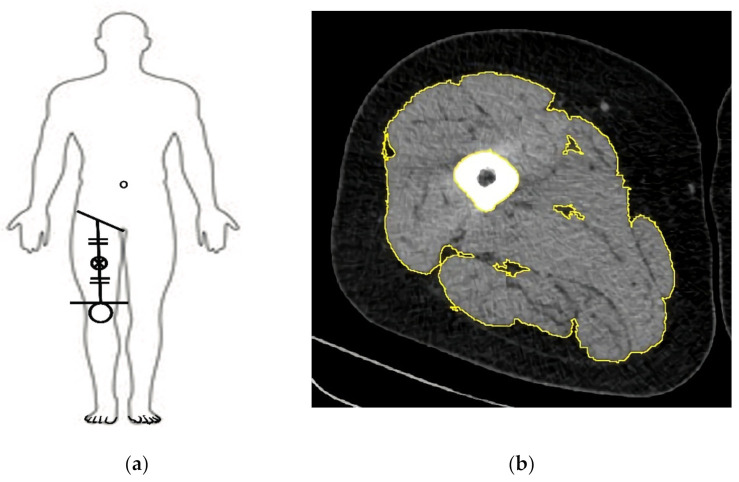
Cross-sectional muscle area (CSMA) determined by mid-thigh CT. (**a**) CSMA was measured at the midpoint from the inguinal crease to the proximal pole of the patella; (**b**) CSA of subcutaneous fat, intermuscular fat, and bone have been removed.

**Table 1 healthcare-09-00711-t001:** Physical characteristics of subjects.

Group	Age (Years)	Height (cm)	Weight (kg)	%Fat (%)	SMM (kg)
SEG (n = 9)	64.8 ± 3.80	153.9 ± 4.22	59.7 ± 6.12	33.6 ± 5.01	20.6 ± 1.74
OSEG (n = 10)	66.6 ± 3.98	151.6 ± 8.61	58.0 ± 11.53	35.8 ± 5.88	20.3 ± 4.41

Values are reported as mean ± standard deviation (SD). SEG, sarcopenia exercise group; OSEG, osteoarthritis sarcopenia exercise group; SMM, skeletal muscle mass.

**Table 2 healthcare-09-00711-t002:** The 15-week complex exercise program for elderly woman with sarcopenia and osteoarthritis.

Phase	Type	Exercise Contents	Time	Intensity/Frequency
AP (1–5 weeks)	Warm-up	Dynamic stretching	10 min	5 days/week
	Work out	Mat exercise	30 min	3 days/week
		Chair exercise		
		Aerobic (walking)	30 min	40–50% HRR (RPE 11–12), 5 days/week
	Cool down	Static stretching	10 min	5 days/week
IP (6–10 weeks)	Warm-up	Dynamic stretching	10 min	5 days/week
	Work out	Mat & gym ball exercise	30 min	3 days/week
		Aerobic (walking)	30 min	50–60% HRR (RPE 12–13), 5 days/week
	Cool down	Static stretching	10 min	5 days/week
MP (11–15 weeks)	Warm-up	Dynamic stretching	10 min	5 days/week
	Work out	Mat & gym ball exercise	30 min	3 days/week
		Aerobic (walking)	40 min	50–60% HRR (RPE 12–14), 5 days/week
	Cool down	Static stretching	10 min	5 days/week

AP, adaptation phase; IP, improvement phase; MP, maintenance phase.

**Table 3 healthcare-09-00711-t003:** Details of the mat, chair, and gym ball exercises of the 15-week complex exercise program.

Phase	Type	Exercise Contents	Repetition	Set & Rest
AP (1–5 weeks)	Mat	SLR (4 directions), prone lat. raise, cat & camel, supine lower trunk rot., wall squat	5–7 reps /3–5 s hold	2–3 sets (1 min rest after set end)
		Supine pelvic tilt	10–15 reps /5 s hold	
		Semi-curl up	10–15 reps	
	Chair	Knee EXT. (alternated), hip FLX./knee EXT. (alternated), knee diagonal EXT. (alternated), toe touch floor (alternated), toe touch floor with reaching out hand (alternated), hip abd./add., hip abd./add. with shoulder ext./int. rot., hip abd./add. with shoulder diagonal open arms, hip abd./add. with touching the head, toe & heel touch floor with walking in place (alternated), hip & shoulder abd./add. with walking in place (alternated)	10–15 reps /activity (with 4/4 beat music)/2 set	1 min rest after 2 sets
IP (6–10 weeks)	Mat	Lower abdominal, prone lying, kneeling hip EXT., kneeling lower trunk rot., wall squat with mini ball, fwd. lunge, kneeling bird dog, lower trunk crawling position	7–10 reps /3–5 s hold	2–3 sets (1 min rest after set end)
		Wall resisting calf raise	10–15 reps /5 s hold	
		Semi-curl up	15–20 reps	
	Gym ball	Heel & toe raise, pelvic tilting (fwd-bwd/lat.)	15–20 reps	2–3 sets (1 min rest after set end)
		Knee EXT., unilat. shoulder FLX. in quadriped bouncing on gym ball, trunk diagonal rot., trunk EXT. in quadriped, unilat. hip FLX. sit on gym ball	7–10 reps /3–5 s hold	
		Write alphabet with foot (A–Z) alternated		
MP (11–15 weeks)	Mat	Fwd. & bwd. lunge, single-side bird-dog, standing on one leg, supine bridge with one leg EXT., prone plank, side plank with knee bent	10 reps /3–5 s hold	2–3 sets (1 min rest after set end)
		Wall resisting calf raise	15 reps /3–5 s hold	
	Gym ball	Hip EXT. with ball under knees, supine knee EXT./FLX., supine int./ext. rot. of leg, hip FLX. in bridge position, knee EXT. in bridge position, hip EXT. with feet on ball	7–10 reps /3–5 s hold	
		Reclined mini curl-up on gym ball	15–20 reps	

AP, adaptation phase; IP, improvement phase; MP, maintenance phase; SLR, straight leg raise; EXT., extension; FLX., flexion; lat., lateral; abd., abduction; add., adduction; rot., rotation; int., internal; ext., external; fwd., forward; bwd., backward’ unilat., unilateral.

**Table 4 healthcare-09-00711-t004:** Serum myokine analysis.

Variables	Group	Pre-Intervention	Post-Intervention	Δ-Value	*t*-Value
IGF-1 (ng/mL)	OSEG	111.17 ± 28.18	129.64 ± 28.08	18.5	−4.74 ***
	SEG	106.99 ± 20.99	107.90 ± 14.09	0.19	−0.89
	*t*-value	−0.36	−2.13	Group × Time F(1,17) = 12.22 **	
Irisin (ng/mL)	OSEG	1.69 ± 0.18	2.38 ± 0.38	0.69	−6.99 ***
	SEG	1.86 ± 0.40	2.64 ± 0.79	0.78	−4.15 **
	*t*-value	−1.20	−0.93	Group × Time F(1,17) = 0.20	
Mstn (pg/mL)	OSEG	1105.22 ± 277.29	775.60 ± 183.35	−329.61	5.45 ***
	SEG	1006.19 ± 234.35	828.72 ± 221.07	−177.46	3.30 **
	*t*-value	0.84	−0.57	Group × Time F(1,17) = 3.47	
IL-10 (pg/mL)	OSEG	101.93 ± 40.83	232.20 ± 74.85	130.26	−7.43 **
	SEG	166.68 ± 58.71	234.21 ± 54.34	67.52	−3.58 **
	*t*-value	−2.82 *	−0.07	Group × Time F(1,17) = 5.96 *	
TNF-α (pg/mL)	OSEG	131.77 ± 6.30	119.95 ± 7.49	−11.82	4.44 **
	SEG	129.72 ± 4.33	124.59 ± 2.26	−5.13	3.56 **
	*t*-value	0.82	−1.78	Group × Time F (1,17) = 4.59 *	

Values are reported as mean ± SD. OSEG, osteoarthritis sarcopenia exercise group; SEG, sarcopenia exercise group; IGF-1, insulin-like growth factor 1; Mstn, myostatin; TNF-α, tumor necrosis factor alpha; SD, standard deviation. * *p* < 0.05. ** *p* < 0.01. *** *p* < 0.001.

**Table 5 healthcare-09-00711-t005:** Cross-sectional area and physical activity ability analysis.

Factor	Group	Pre	Post	Δ-Value	*t*-Value
L-TCSA (cm²)	OSEG	88.48 ± 15.19	94.52 ± 14.70	6.04	−8.23 ***
	SEG	99.62 ± 6.91	110.42 ± 6.04	10.80	−6.36 ***
	*t*-value	−2.02	−3.02 **	Group × Time F(1,17) = 7.12 *	
R-TCSA (cm²)	OSEG	89.36 ± 14.26	93.06 ± 16.36	3.70	−3.27 **
	SEG	102.44 ± 11.21	110.04 ± 6.98	7.61	−3.31 *
	*t*-value	−2.20	−2.88 **	Group × Time F(1,17) = 2.5	
WOMAC (score)	OSEG	61.70 ± 17.493	40.30 ± 13.132	−21.40	6.24 ***
	SEG	43.22 ± 13.122	35.22 ± 11.278	−8.00	1.99
	*t*-value	1.77	−0.52	Group × Time F(1,17) = 5.83 *	
SPPB (score)	OSEG	6.70 ± 1.160	12.10 ± 0.876	5.4	−17.68 ***
	SEG	8.67 ± 1.500	12.44 ± 1.130	3.78	−5.89 **
	*t*-value	−3.21 **	−0.74	Group × Time F(1,17) = 5.58 *	

Values are reported as mean ± SD. OSEG, osteoarthritis sarcopenia exercise group; SEG, sarcopenia exercise group; L-TCSA, left thigh muscle cross-sectional area; R-TCSA, right thigh muscle cross-sectional area; WOMAC, The Western Ontario and McMaster Universities Osteoarthritis Index; SPPB, short physical performance battery; SD, standard deviation. * *p* < 0.05. ** *p* < 0.01. *** *p* < 0.001.

## Data Availability

The data presented in this study are available on request from the corresponding author. The data are not publicly available due to privacy.

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
