# Peer review of "Complex Exercise Improves Anti-Inflammatory and Anabolic Effects in Osteoarthritis-Induced Sarcopenia in Elderly Women"

_healthcare, 2021, doi:10.3390/healthcare9060711_

Round 1

Reviewer 1 Report

Congratulations on your work. I attach a series of suggestions:

  1. Inclusion criteria should include whether information was provided to the participants about the study, as well as whether they signed a consent to participate.
  2. On the other hand, it would have been advisable to include a control group without exercise or with exclusively aerobic work. Could you justify the absence of a control group?
  3. Indicate which professionals or professional category carried out both the measurements and the training program (doctors, experts in physical activity and sport, etc.)
  4. You provide a good explanation about the structure of the sessions, however, the description of some aspects is missing, such as if a day of rest was left between sessions or if there were variations in the sessions of the same week, that is, Did it take performed the same session five days a week, for 5 weeks?
  5. Also, should they indicate if the sessions were group or individual, were led by a coach or autonomous, or were they held in a specific facility? I advise you to include the information related to these concepts.
  6. Explain if you had any limitations during the study.

Author Response

Thanks for the good points and advice.

Reviewer 2 Report

Dear authors, in the attached document you can find the considerations and suggestions for improving your work.
Best regards.

Author Response

Thanks for the good points and advice.
